# Integration of Capacitive Pressure Sensor-on-Chip with Lead-Free Perovskite Solar Cells for Continuous Health Monitoring

**DOI:** 10.3390/mi14091676

**Published:** 2023-08-27

**Authors:** Sameh O. Abdellatif, Afaf Moustafa, Ahmed Khalid, Rami Ghannam

**Affiliations:** 1The Electrical Engineering Department, Faculty of Engineering and FabLab, Centre for Emerging Learning Technologies (CELT), The British University in Egypt (BUE), Cairo 11387, Egypt; sameh.osama@bue.edu.eg (S.O.A.); afaf182963@outlook.com (A.M.); ahmed197380@bue.edu.eg (A.K.); 2James Watt School of Engineering, University of Glasgow, Glasgow G12 8QQ, UK

**Keywords:** finite element analysis, circuit layout, lead-free perovskite, power management, flexible electronics

## Abstract

The increasing prevalence of hypertension necessitates continuous blood pressure monitoring. This can be safely and painlessly achieved using non-invasive wearable electronic devices. However, the integration of analog, digital, and power electronics into a single system poses significant challenges. Therefore, we demonstrated a comprehensive multi-scale simulation of a sensor-on-chip that was based on a capacitive pressure sensor. Two analog interfacing circuits were proposed for a full-scale operation ranging from 0 V to 5 V, enabling efficient digital data processing. We also demonstrated the integration of lead-free perovskite solar cells as a mechanism for self-powering the sensor. The proposed system exhibits varying sensitivity from 1.4 × 10^−3^ to 0.095 (kPa)^−1^, depending on the pressure range of measurement. In the most optimal configuration, the system consumed 50.5 mW, encompassing a 6.487 mm^2^ area for the perovskite cell and a CMOS layout area of 1.78 × 1.232 mm^2^. These results underline the potential for such sensor-on-chip designs in future wearable health-monitoring technologies. Overall, this paper contributes to the field of wearable health-monitoring technologies by presenting a novel approach to self-powered blood pressure monitoring through the integration of capacitive pressure sensors, analog interfacing circuits, and lead-free perovskite solar cells.

## 1. Introduction

The phrase “wearable capacitive pressure sensors” refers to using capacitive pressure sensors in wearable technology to track various physiological and physical aspects [1,2,3,4]. These sensors help track many bodily and physical elements [1,2,3,4]. They can monitor important bodily data such as heartbeats, blood volume, and breathing rates, making them very useful in wearable technology [5]. Capacitive pressure sensors are often chosen for wearable technology since they offer many benefits over other types of sensors [2]. These good points include being small in size, which is very important for wearable technology where weight and size are key parts of the design [6,7]. Low power consumption is another edge for capacitive pressure sensors [8,9].

Additionally, the high accuracy of capacitive pressure sensors is crucial for applications requiring accurate pressure measurements. In addition, capacitive pressure sensors are impervious to electromagnetic interference, making them perfect for wearable technology exposed to high amounts of electromagnetic radiation [10], such as that used in medical applications. Lastly, durability and capacitive pressure sensors resist harsh environmental conditions like high degrees of vibration and varying temperatures [11]. Therefore, they are perfect for wearable technology usage for arduous environments like sports or military applications [9].

Recent developments in capacitive pressure sensor technology have the potential to impact the creation of wearable technologies substantially. Among the significant developments are new materials where the production of more adaptable and robust capacitive pressure sensors has been made possible by developing new materials, such as conductive polymer composites [2,3]. This is crucial for wearable technology since the sensors must be strong enough to sustain regular use. Moreover, the new capacitive pressure sensors that are smaller and more flexible have been made possible by advancements in fabrication techniques [12,13,14], including micro-fabrication and printed electronics [15,16]. As a result, sensors may now be integrated into wearable technology in novel and creative ways. Again, new integration methods have made it possible to integrate capacitive pressure sensors seamlessly and comfortably into wearable devices, such as wearable patches or textiles [17,18]. Likewise, with enhanced algorithms, the accuracy and dependability of capacitive pressure sensor measurements have increased.

Although capacitive pressure sensors recorded booming performance as a robust wearable sensor, there are still some challenges to on-chip wireless sensor nodes [19]. Firstly, the effects of motion and body posture. The wearer’s motion and body position can impact the capacitive pressure sensors’ accuracy [5]. For instance, the pressure sensor may produce erroneous indications when the wearer is moving since the motion causes changes in pressure. Another aspect is skin contact [20,21]. Because capacitive pressure sensors measure pressure by skin contact, it is critical to ensure the sensor is positioned correctly on the skin to avoid inaccuracies. The accuracy of the pressure measurement can also be impacted by the characteristics of the skin, such as wetness, oil, and hair, as well as variability in temperature [19]. Capacitive pressure sensors are susceptible to temperature variations, which may result in inaccurate measurements. Temperature compensation methods are employed to account for temperature variations to solve this problem [9,22,23].

Other challenges can occur from the system’s point of view. As the sensing research community is pursuing innovative sensor-on-chip technology, sensing devices are not the only challenge in design and fabrication. Sensors are back-staged by an analog interfacing circuit that converts the sensing signal into an appropriate voltage level to be digitally processed through an analog-to-digital converter (ADC). A communication link transceiver is also added to the chip (Figure 1) for wireless connectivity. Finally, self-powered sensors are operated with an energy harvesting unit backed up by a storage buffer. Enlarging the sensing technology from a simple passive device to a sophisticated smart sensing unit with energy and wireless communication electronics can be seen as a revolution in the sensor industry. However, such enlargement increases the design challenges exponentially in terms of energy management, CMOS technology computability, and limited area restrictions [24].

In the current study, we demonstrated a complete attempt to simulate a smart sensor-on-chip capacitive pressure sensor for wearable applications. Experimentally verified finite element analysis for a capacitive pressure sensor was modeled using COMSOL Multiphysics. An analog circuit then interfaced the variation in capacitance to feed an ADC. Herein, two schematics were proposed to discuss various trade-offs, evaluating parameters such as energy consumption and layout area needed. Finally, the energy budget is investigated using a lead-free perovskite low-light energy harvester. Overall, system integration and energy management are presented as well.

The novelty of this work lies in the comprehensive multi-scale simulation and integration of various cutting-edge technologies in the development of a self-powered blood-pressure capacitive pressure wearable sensor-on-chip. Firstly, the integration of analog, digital, and power electronics into a single system presents a significant challenge, and this paper addresses this challenge by proposing two analog interfacing circuits that enable efficient digital data processing. Secondly, the integration of lead-free perovskite solar cells as a mechanism for self-powering the sensor is a unique approach that enhances the device’s sustainability and eliminates the need for external power sources. This integration of perovskite solar cells as an energy harvesting mechanism showcases the potential for sustainable power generation in wearable devices. Additionally, the varying sensitivity of the proposed system based on the pressure range of measurement adds versatility to the device, allowing it to cater to different monitoring needs. To minimize the off-chip area and optimize the overall design, we aimed to integrate both the capacitive sensor and the perovskite lead-free light harvester into a single indium-tin-oxide (ITO)-coated flexible polyethylene terephthalate (PET) substrate, as illustrated in Figure 1. The capacitive pressure sensor senses the applied pressure and converts it into variations in capacitance. Simultaneously, the perovskite cell harvests light energy to power the entire chip, including the capacitive sensor. This integrated approach ensures efficient utilization of the available space and enables seamless operation of both the sensor and the harvester. By combining these functionalities, we can achieve a compact and self-sustaining system that effectively captures pressure variations and utilizes harvested light energy for power generation.

## 2. Capacitive Pressure Sensor

In this research, we utilized a flexible capacitive pressure sensor, featuring an active layer of porous polydimethylsiloxane (PDMS) sandwiched between two electrodes of indium-tin-oxide (ITO)-coated flexible polyethylene terephthalate (PET) substrate, as illustrated in Figure 1 and Figure 2. The sensor structure, material, and pressure sensing range were chosen according to the work in [5] for experimental validation. The work in [5] presented several key advantages. Firstly, the sensor utilized a low-cost PDMS-DI water dielectric, making it flexible and wearable for comfortable and practical use. Moreover, the PDMS-DI substrate can adapt the lead-free perovskite solar cell (see Figure 1), seeking self-powered sensor. Secondly, it exhibited a relative difference in capacitance of 0.07–15% across a wide pressure range from 1 Pa to 100 kPa, indicating its high sensitivity and accuracy. Thirdly, the sensor demonstrated a high working stability, with a quick response time of approximately 110 ms and an ultra-low detection limit of pressure at 1 Pa.

The porosity of the PDMS was modeled using the effective medium theory, as in our previous work in [25]. The proposed sensing device was simulated using COMSOL Multiphysics as a blood pressure sensor. COMSOL Multiphysics was used for its wide applicability in simulating various physical phenomena, its capabilities in handling multi-physics interactions, and its flexibility in defining custom physics interfaces. Given the complex nature of the capacitive pressure sensor, which involves simultaneous mechanical and electrical interactions, COMSOL offers the necessary tools and versatility to accurately model and analyze the sensor’s behavior.

To simulate a capacitive pressure sensor using COMSOL, the boundary conditions were set to free boundaries, and the applied pressures that were relevant to the simulation were inputted. The simulation was then performed by solving the governing equations for the piezo-effect. This typically involves solving the coupled equations for the mechanical deformation and the resulting electrical response. Herein, we utilized the solid mechanics and electrostatics modules. The material parameters for the PDMS have been extracted from [26,27,28]. The counter electrode was considered isolated to any pressure, while the front electrode was pressured with a staircase variation from 1 Pa to 100 kPa.

The variation in the piezo-effect against the pressure range was modeled in Figure 2a–e. The relative difference in capacitance for the applied pressure range from 1 Pa to 100 kPa recorded was from 0.07 to 15.4%. The pressure was demonstrated in the log scale to present the wide range of pressure variation across five orders of magnitudes. Herein, the initial capacitance $C_{0}$ is 6.14 pF; more details are addressed in the next section. The sensuosity of the proposed sensor was detected as the slope of the data given in Figure 2, as:(1)$$S=\frac{\Delta C/C_{0}}{\log\Delta (P)}$$
where $\Delta C/C_{0}$ is the relative difference in capacitance, and P is the applied pressure in Pa. Generally, the proposed sensing range can be divided into sensing ranges, from 1 Pa to 10 Pa, from 10 Pa to 100 Pa, from 100 Pa to 1 kPa, and from 1 kPa to 100 KPa. The sensor sensitivity showed variation around 1.7 × 10^−3^ (kPa)^−1^, 0.068 (kPa)^−1^, 0.095 (kPa)^−1^, and 1.4 × 10^−3^ (kPa)^−1^. The simulated results reported in this work are experimentally verified with the previous data in the literature given in [5]. The root-mean-square (RMS) error between the 16 experimental points, demonstrated in Figure 2a, and the corresponding simulation results was 7.37%. In the next section, the capacitance variation is interfaced to be converted into voltage variation. As the target is a CMOS-based sensor-on-chip, the capacitive pressure sensor is treated as off-chip capacitance. In contrast, the other capacitors are chosen to be in the pF range to fit with the 28 nm utilized CMOS technology.

## 3. Interfacing Circuit

Following the capacitive pressure sensor demonstrated in Section 2, the variation in the pressure is electrically converted as a variation in capacitance. However, to digitally analyze an electric signal, voltage variation is required. Accordingly, the back-stage analog interface circuit is designed for that purpose. Referring to the literature, capacitive interfacing circuits can be classified into three categories: capacitance-to-frequency circuits, switching capacitors with amplifiers, and capacitance-to-voltage converters [29]. Herein, we chose the capacitance-to-voltage converters as a low-power and simple alternative [29,30]. We propose two schematics to act as capacitive sensor interfacing circuits (see Figure 3). Generally, the piezo-capacitor sensor is connected to a 5-pF capacitor, where the primary voltage driving point is the intermediate node. The first proposed circuit (Figure 3a) shows a simple double-stage interfacing circuit with a buffer, followed by the non-inverting amplifier. Alternatively, the circuit in Figure 3b demonstrated a multi-staged interfacing circuit. The multi-staged circuit is separated by MOSFETs acting as switches with controlled triggering signals through the transistor gate. The multi-stage circuit is designed sequentially to process the sensed signal.

Initially, the signal is filtered with a first-order filter that decays sharp high-frequency responses. A standard buffer is then used with a second-order active filter with gain. Finally, the non-inverting amplifier is added to tune the output voltage range. For comparison, the two circuit’s output voltage is adjusted to vary from 0 V to 5 V, as seen in Figure 4. The results in Figure 4 were obtained via the circuit schematics in Figure 3 using Multisim, where the $C_{1}$ capacitance was varied to simulate the variation in the capacitive pressure sensor. The multi-stage interfacing circuit showed better performance as a linear C-V characteristic curve. We attribute this to the double filtering processes that narrow the root-mean-square error (RMS) against the linear fitting. The proposed circuits showed transfer sensitivity of 0.023 mV/pF ± 9.4% and 0.023 mV/pF ± 3.8%.

Consequently, the driving interfacing circuits were scaled down to the transistor level, up to a layout in Figure 5. As illustrated earlier, one of the main challenges of the current study is to propose an efficient sensor-on-chip using CMOS technology with minimum off-chip components. Herein, we utilize the 28 nm CMOS technology to layout the interfacing circuits. The secondary proposed interfacing circuit, the multi-stage shown in Figure 3b, includes two off-chip capacitors. The output terminals of the driving chip can supply a voltage range from 0.1 × $V_{CC}$ up to 0.96 × $V_{CC}$. This results in a voltage range from 50 mV to 4.6 V. Typically, the rated output voltage range is chosen to adopt a wide range of IoT applications. In this study, predominantly in Section 5, the two circuits are compared in terms of power and energy consumption as well as the required layout area. The layout presents two single-chips with an area of 0.785 × 0.757 mm^2^ and 1.78 × 1.232 mm^2^ under the 28 nm low voltage (L.V.) CMOS technology (cf. Figure 6). The layout was developed using L.V. MOSFETs, which were acting as switches.

## 4. Low-Light Energy Harvesting

Wearable electronics traditionally depend on DC batteries. However, this approach confines the system’s lifespan to that of the battery. As such, the concept of “self-powered sensors” emerges when alternative power sources are used to supplement or replace the battery [20,31,32,33]. Among various alternatives, light harvesters can be treated as one of the solutions capable of being integrated into wearable electronics [1]. The idea is illustrated in the possibility of having a light harvester fabricated on a flexible substrate, mostly the same sensor substrate [34,35,36]. However, using such photovoltaics is still restricted by some challenges, including but not limited to the limited area and the operation under indirect light. Concerning indirect light operation, or what we can call operation under diffused light, perovskite solar cells (PSC) have recorded significant contributions concerning conventional silicon cells [37]. This promotes perovskite solar cells for such flexible electronic circuits. Lead-free perovskite solar cells are highly recommended, especially with biosensing applications [20,38,39,40,41].

We used a lead-free perovskite solar cell in this research, following our previous work in [20]. The simulated cell was based on the following architecture: ITO-PET/PC_61_BM/MASnI_3_/PEDOT: PSS + ITO-PET. It can be observed that the sympatric cell is typically designed with the same substrate used for the capacitive pressure sensor, ITO-PET. The fullerene derivative [6,6-phenyl-C_61_ butyric acid methyl ester (PCBM), acting as an electron transport layer, is a solubilized version of the buckminsterfullerene. Following the data reported in the literature, PCBM enables rapid and efficient charge transfer and exaction dissociation and has a high electron mobility [42]. As a perovskite layer, methylammonium tin iodide MASnI_3_ is classified as a promising lead-free perovskite for optoelectronic applications [43]. Finally, poly(3,4-ethylene-dioxythiophene) polystyrene sulfonate (PEDOT: PSS) was selected as a hole transport layer [44,45]. As stated earlier, skin contact can be one of the factors affecting the performance of the wearable sensor. In this context, we modeled the skin as a counter layer, using the optoelectronic skin model introduced in [20,46,47].

As a result, the J-V characteristic curve for the proposed lead-free cell was simulated as shown in Figure 6. We used our previous perovskite solar cell modeling methodology that relies on the Solar Cell Capacitance Simulator (SCAPS-1D) program, as reported in [48,49]. The main theory of operation for the SCAPS-1D is based on the simulation of the electrical characteristics of solar cells, specifically their capacitance behavior. SCAPS-1D is a numerical simulation tool that uses a one-dimensional approach to model the physical and electrical properties of solar cells. The simulator considers the various layers and interfaces within a solar cell structure, including the semiconductor materials, metal contacts, and insulating layers. It considers the effects of doping concentrations, bandgap energy, carrier mobility, and other material parameters that influence the device’s performance. SCAPS-1D utilizes the drift-diffusion equations and Poisson’s equation to model the carrier transport and charge distribution within the solar cell. It simulates the generation, recombination, and collection of electron-hole pairs under different illumination conditions.

Standardly, the cell’s performance under the one-sun AM1.5G spectrum was modeled, recording open-circuit voltage ($V_{OC}$) around 0.756 V with a short-circuit current density ($J_{SC}$) of 23.13 mA/cm^2^. The overall power conversion efficiency ($\eta_{PCE})$ showed 7.87%. Based on the nature of the applications, the wearable flexible substrate used in such a sensing technology is far from working under standard direct light operation. Consequently, we proposed the operation under two other harsh conditions. The first is a diffused light operation, where diffused light is defined here as the light beams with an angle of incidence greater than 45°, following the diffused light ratio defined in [25]. The recorded data, under diffused light with a nearly 40% reduction in $J_{SC}$ and $\eta_{PCE}$ of 5.17%, promote a selling edge for these lead-free perovskite solar cells. Referring to silicon, the diffused light ratio ($D_{45{}^{\circ}}$) in PSC was found to be 62.1%, while it recorded 32.9% in silicon, as in [25]. This illustrates the superiority of the PSC on Si-based cells under diffused light. Another harsh condition can be demonstrated as a direct beam with low-intensity spectra. Herein, we consider the low-light intensity as 0.2 sun [25,48,50]. The observed characteristic shown in Figure 6 indicates 20% of the $J_{SC}$ for the one-sun condition, as expected. The main output characteristic parameters for the lead-free cell under the various simulation conditions are listed in Table 1.

## 5. System Integration

This section demonstrates the overall system integration with the associated power management process. Following the block diagram introduced in Figure 1, the current study considered an already made 5-V ADC unit, a 433 MHz transceiver (reported in [51,52], a lithium cell 3-V battery, and a 0.5–3 V boost converter. As observed earlier, one of the advantages of capacitive pressure sensors is the ultra-low power consumption. Accordingly, the interfacing circuit is considered the main power drain in this sensor. The overall power consumption for the two proposed interfacing circuits is listed in Table 2, along with the overall system specification. It can be detected that the multi-stage circuit consumes less power than the double-stage circuit. This reflects a 30% reduction in the PSC area needs for light harvesting, assuming a dominating diffused spectrum.

Additionally, as mentioned in Section 3, the multi-stage filtering circuits reject high-frequency components, treated as noise for this system. This leads to a minimal RMS error concerning the expected linear transfer characteristics. The main drawback for the multi-stage interfacing circuit, concerning the double-stage circuit, is the need for off-chip capacitance in performing the inductor filtering stages.

## 6. Conclusions

The primary objective of this paper is to present a comprehensive system model for a blood-pressure capacitive sensor that includes an active layer of porous polydimethylsiloxane (PDMS) sandwiched by two electrodes of indium-tin-oxide (ITO) coated on a flexible polyethylene terephthalate (PET) substrate. The presented comprehensive model of a blood-pressure capacitive sensor showcases promising possibilities for continuous, non-invasive health monitoring, addressing the persistent challenge of hypertension. The sensor is back-staged with an analog interfacing circuit, where two alternatives have been proposed. The circuits showed 0.023 mV/pF ± 9.4% C-V transfer characteristics for a double-stage circuit and 0.023. mV/pF ± 3.8% for the multi-stage circuit. The power management was introduced with a diffused light energy harvester, simulated using a lead-free perovskite solar cell. The cell efficiency under diffused light was recorded at 5.17%, with an enhanced diffused light ratio compared to traditional silicon cells. The multi-stage interfacing circuit showed 30% lower power consumption with a relatively higher layout area needed. Moreover, the introduction of a diffused light energy harvester, particularly a lead-free perovskite solar cell, to power the sensor represents a step forward in achieving sustainability in wearable technology. Its improved efficiency under diffused light, compared to traditional silicon cells, increases the sensor’s adaptability to different lighting conditions and locations, thereby widening its potential applications. As part of our future work, a physical prototype utilizing CMOS chip technology is envisioned. In this prototype, integrating the sensor and the harvester as off-chip components will be considered. By adopting this approach, we aim to enhance the overall functionality and performance of the system. This will involve careful consideration of the design and integration aspects to ensure seamless operation and optimal efficiency.

## Figures and Tables

**Figure 1 micromachines-14-01676-f001:**
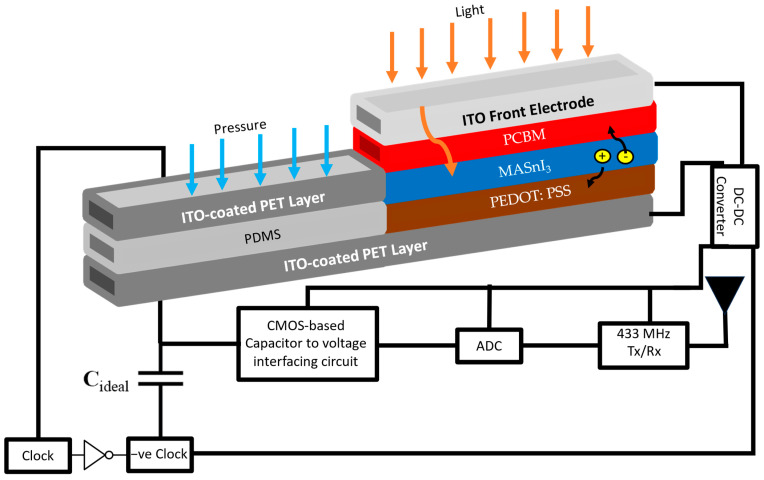
The macroscopic block diagram for the smart capacitive pressure sensor-on-chip.

**Figure 2 micromachines-14-01676-f002:**
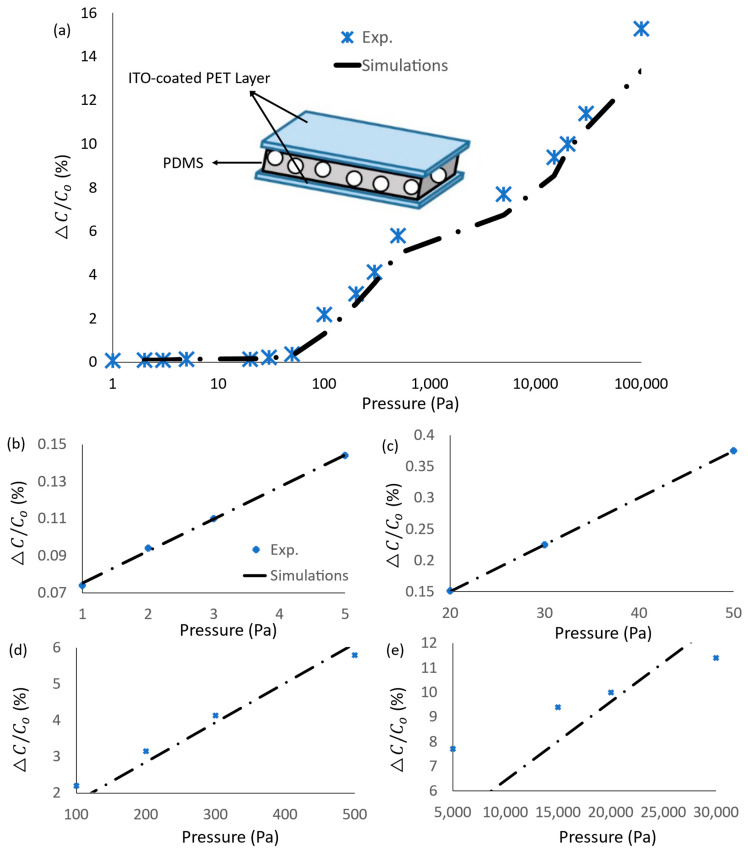
Capacitive pressure sensor; pressure variation against incremental variation in the capacitance. Experimental data are captured from [5]: (**a**) The fill pressure range in log-scale, (**b**) linear scale from 1 Pa to 5 Pa, (**c**) linear scale from 20 Pa to 50 Pa, (**d**) linear scale from 100 Pa to 500 Pa, and (**e**) linear scale from 5 KPa to 30 KPa.

**Figure 3 micromachines-14-01676-f003:**
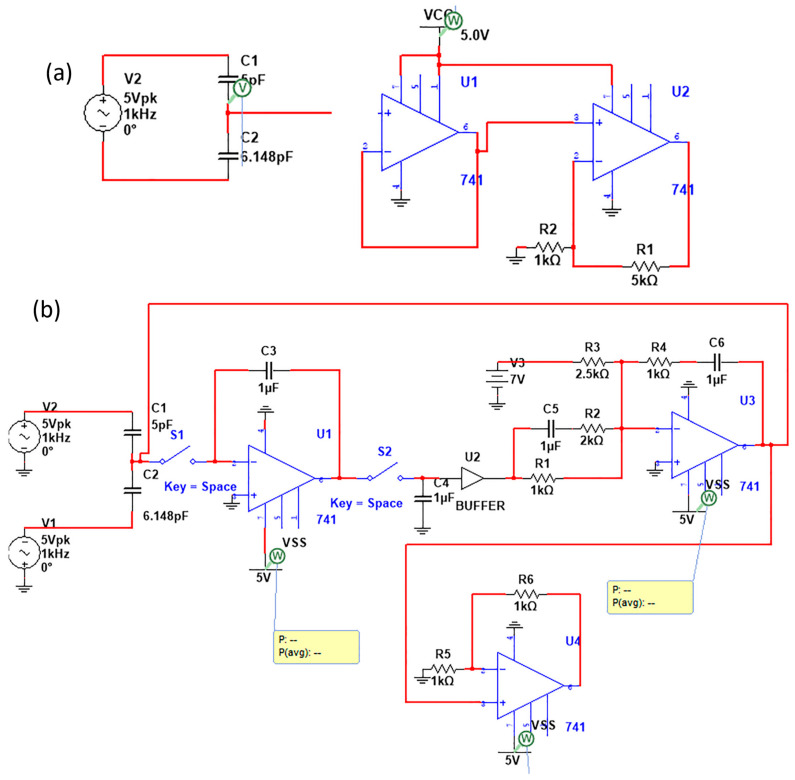
Capacitive sensor interfacing circuit: (**a**) simple double-stage circuit and (**b**) multi-stage capacitor-to-voltage converter.

**Figure 4 micromachines-14-01676-f004:**
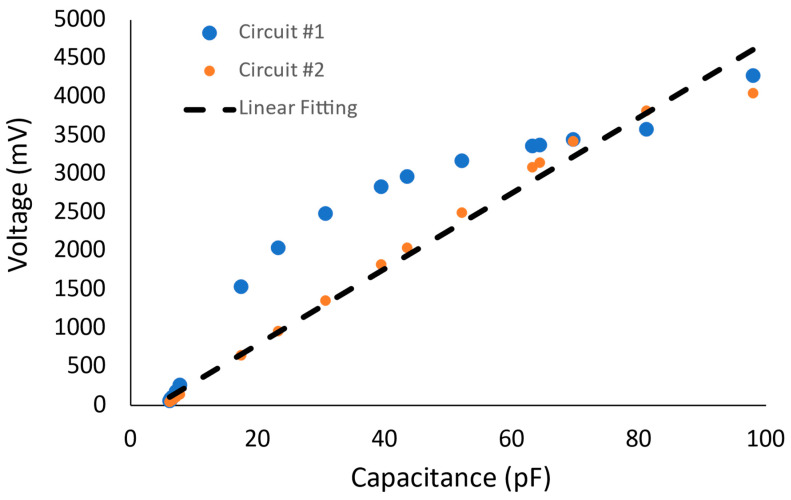
Capacitive sensor interfacing circuit voltage-capacitance characteristic curve for both simple double-stage circuit and the multi-stage capacitor-to-voltage converter; agonist a linear fitting curve.

**Figure 5 micromachines-14-01676-f005:**
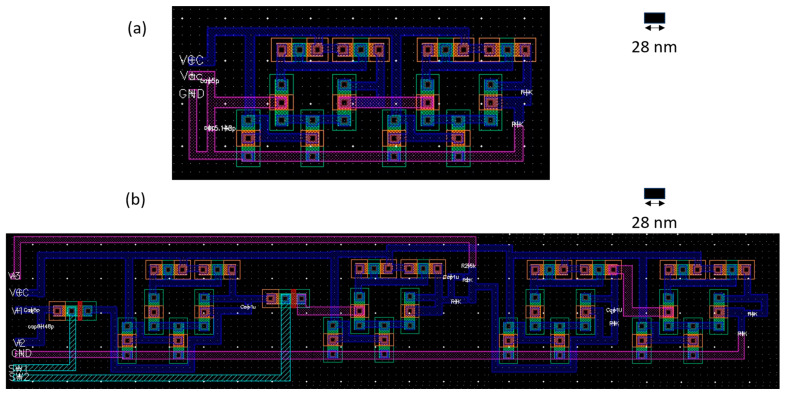
Capacitive sensor interfacing circuit layout for (**a**) simple double-stage circuit (Circuit #1 in Figure 4) and (**b**) multi-stage capacitor-to-voltage converter (Circuit #2 in Figure 4).

**Figure 6 micromachines-14-01676-f006:**
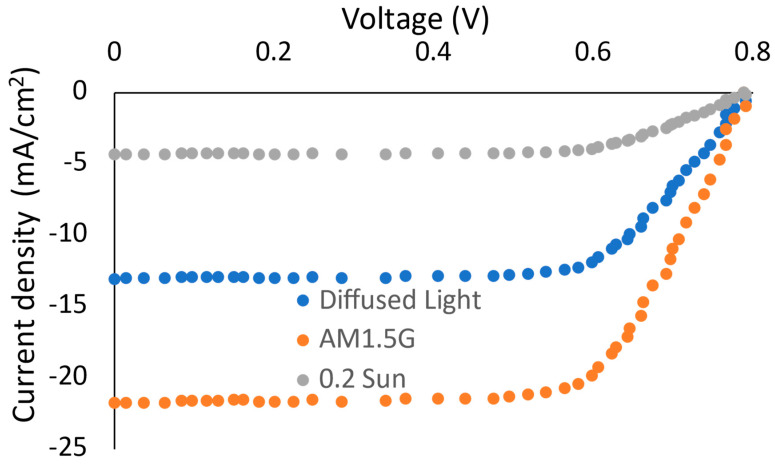
The J-V characteristic curves for lead-free perovskite solar cell under one-sun spectrum, diffused light and 0.2 sun.

**Table 1 micromachines-14-01676-t001:** The main output characteristic parameters for the lead-free cell under various optical simulation conditions.

Optical Condition	Short-Circuit Current Density ($J_{SC}$)	Open-Circuit Voltage ($V_{OC}$)	Fill Factor	Power Conversion Efficiency ($\eta_{PCE}$)
AM1.5G	23.13 mA/cm^2^	0.756 V	71.14%	7.87%
Diffused light	13.25 mA/cm^2^	0.741 V	69.72%	5.17%
0.2 Sun	4.78 mA/cm^2^	0.737 V	70.55%	7.68%

**Table 2 micromachines-14-01676-t002:** Sensor-on-chip overall specification using the double-stage and the multi-stage interfacing circuits.

System Parameter	Double-Stage Circuit	Multi-Stage Circuit
Sensor sensitivity (1 Pa to 10 Pa)	1.7 × 10^−3^ (kPa)^−1^	
Sensor sensitivity (10 Pa to 100 Pa)	0.068 (kPa)^−1^	
Sensor sensitivity (100 Pa to 1 kPa)	0.095 (kPa)^−1^	
Sensor sensitivity (1 kPa to 100 kPa)	1.4 × 10^−3^ (kPa)^−1^	
$\Delta C/C_{0}$	0.07–15.4%	
ΔC	6.15 pF to 98.02 pF	
Output Voltage	50 mV to 4.6 V	
The circuits transfer sensitivity	0.023 mV/pF ± 9.4%	0.023 mV/pF ± 3.8%
The layout area	0.785 × 0.757 mm^2^	1.786 × 1.232 mm^2^
Power consumed	72.7 mW	50.5 mW
PSC required area	9.76 mm^2^	6.487 mm^2^

## Data Availability

The data that support the findings of this study are available from the first author upon reasonable request.

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
