# Peer review of "Integration of Capacitive Pressure Sensor-on-Chip with Lead-Free Perovskite Solar Cells for Continuous Health Monitoring"

_micromachines, 2023, doi:10.3390/mi14091676_

Round 1

Reviewer 1 Report

In this work, the authors demonstrated a comprehensive multi-scale simulation of a sensor-on chip that is based on a piezo-capacitive pressure sensor. Their findings underline the potential for such sensor-on-chip designs in future wearable health-monitoring technologies. Therefore, this work can be accepted after a minor revision. The detailed comments are as follows.

1. The schematic diagrams or photographs of the system including the piezo-capacitive pressure sensor and the perovskite solar cell should be provided.

2. Please cite more references, such as “https://www.nature.com/articles/s41467-022-33454-y”, to improve the strength of the manuscript.

3. The manuscript should be polished, and the various format errors should be revised.

The Quality of the English Language is good.

Author Response

Dear Reviewer 1,

Thank you very much for your kind comments. We are uploading a point-by-point response letter that contains all the necessary revisions.

Regards,

Rami.

Reviewer 2 Report

The manuscript “Simulating Self-powered Blood-pressure Piezo-capacitive Wearable Sensor-on-chip using Lead-free Perovskite Low-light Harvester” simulated the sensitivity of a piezo-capacitive pressure sensor, proposed two analog interface circuits to process the electrical signal for detecting the capacitor change of the sensor and designed COMS layout for these two circuits. The authors also simulated J-V characteristic curves for lead-free perovskite solar cell under under One-Sun AM1.5G, diffused light, and 0.2 Sun lighting conditions. The work attempts to develop a self-powered and sensor-on-chip based blood pressure sensor, however it is at an early stage so the authors report the simulation results only. I would suggest the publication of this manuscript on Micromachines with following major revisions:

1.       Authors should highlight the novelties of this manuscript.

2.       Please explain why the piezo-capacitive sensor based on reference 5 is selected in this work.

3.       Please give out the simulation conditions with COMSOL for the simulation of the piezo-capacitive pressure sensor in Figure 2.

4.       Is it possible to add a curve in linear scale from 10 pa to 1000 pa, and a linear curve from 1 kpa to 100 kpa? Reference 5 is with linear scale.

5.       In line 124, kindly explain how is the RMS calculated? Does it include all data points in simulation and experiments?

6.       For Figure 4 results, are they obtained by simulation or by experiments? If they are acquired by simulation, what is the simulation software and how is it simulated?

7.       In Figure 5 in line 189, please indicate which parts in Figure 5(a) and 5(b) are relating to the 2 circuits in Figure 4. Please add a scale bar for the two layouts.

8.       In Section 4, explain in detail about the simulation conditions when using the software SCAPAS, and do not use a simple sentence “We used our previous perovskite solar cells modeling methodology using SCAPAS, as reported in [48,49]” to omit the details.

9.       Some typos to correct:

·       In line 224, “Figure 5” should be “Figure 6”, “for lead-0free” should be “lead-free”.

·       In line 226, “Table 2” should be “Table 1”.

·       Delete the lines from 282-303.

·       Indicate the full name for PCBM.

·       Indicate the the full name for PSC.

Author Response

Dear Reviewer 2,

Thank you very much for your comments. Please find enclosed a point-point response letter that highlights the necessary revisions to the paper.

Best wishes,

Rami.

Round 2

Reviewer 2 Report

The revised version is much improved and I recommend it to be published on Micromachines as it is. 

Author Response

We thank the reviewer for these valuable comments and suggestions.